# OpenReview forum: "Scaling Long-Horizon Agent via Context Folding"
_ICML.cc/2026/Conference — ICML 2026 regular_

### Official Review · Reviewer_dD92 · 2026-02-20

**Soundness:** 3
**Presentation:** 3
**Significance:** 3
**Originality:** 3
**Overall Recommendation:** 5
**Confidence:** 3

**Summary:**

This paper proposes the FoldGRPO to enable existing LLM agents to handle long context by themselves via an end-to-end RL manner. It allows the agent to tackle subtasks with a branched sub-trajectory and fold it upon its completion to allow a larger context window size. The implementation of such operations is triggerred by the agent itself via `branch` and `return`, which natively generalizes to existing LLM agents with tool-use capabilities. The extension from 32K to 327K context window brings improvement gains on both browsecomp-plus and SWE-Bench.

**Compliance With Llm Reviewing Policy:**

Affirmed.

**Final Justification:**

Recommend accept.

**Key Questions For Authors:**

- The reviewer believes the comparison on single-hop/multi-hop QA benchmarks can be added because they are: 1) easy to implement, and 2) broadly used across methods for long context window training/multiple turns of interactions. The supplement of this experiment can enhance the proposed method for comparability.
- The training of the proposed method might generalize to Qwen/LLaMA series as the Seed-OSS is not popular in research literature and remains unknown for comparison fairness.
- The implementation of the GRPO RL process with changing trajectories (e.g., varied prefix contex) should be fully described in details, since this would affect the optimization of GRPO.
- More training and validation dynamics such as the number of folding actions, the context window size, and the number of samples (not the trajectory-level but the actual samples involved in policy gradient update) are highly encouraged to provide with in-depth discussion.
- The following work on RLVR for agentic scenarios can be considered to add into the related work:
Shi, Yuchen, et al. "Youtu-agent: Scaling agent productivity with automated generation and hybrid policy optimization." arXiv preprint arXiv:2512.24615 (2025).
Qin, Yulei, et al. "Learn the ropes, then trust the wins: self-imitation with progressive exploration for agentic reinforcement learning." arXiv preprint arXiv:2509.22601 (2025).

**Limitations:**

yes

**Strengths And Weaknesses:**

### Summary Of Strengths
- This paper is well-written with beautiful illustrations.
- The method itself is intuitive and the processing of sub-tasks in separate trajectories is effective.

### Summary Of Weaknesses
- The open-sourcing of the FoldGRPO is not mentioned in the paper and the authors are encouraged to make them public to benefit the community.
- The lack of comparison with existing long-context training/agentic scaffolding/agentic RL with frameworks would devalue the proposed method as we only see the vanilla GRPO is involved in Table 1 and 2.

---

> ### Author Rebuttal · Authors · 2026-03-31
>
> Thank you for your valuable comments! We are glad to hear that you find our work well-written and the proposed method effective. We would like to address your concerns as follows:
>
> ---
>
> > **The open-sourcing of the FoldGRPO is not mentioned in the paper and the authors are encouraged to make them public to benefit the community.**
>
> We will open-source our code and data upon publication.
>
> ---
>
> > **The lack of comparison with existing long-context training/agentic scaffolding/agentic RL with frameworks would devalue the proposed method as we only see the vanilla GRPO is involved in Table 1 and 2.**
>
> To clarify, we also compare against a summary-based agent as a major long-context baseline. As discussed in Appendix D and Table 4, we implement a summary agent as an apples-to-apples, well-tuned long-context agentic RL baseline. Our comparison focuses on context management strategies rather than specific RL algorithms (e.g., GRPO vs. PPO), so we keep the same GRPO training framework across methods to ensure a fair comparison and isolate the effect of context management.
>
> ---
>
> > **The reviewer believes the comparison on single-hop/multi-hop QA benchmarks can be added because they are: 1) easy to implement, and 2) broadly used across methods for long context window training/multiple turns of interactions. The supplement of this experiment can enhance the proposed method for comparability.**
>
> We appreciate the suggestion! We evaluated BrowseComp because widely used multi-hop QA datasets such as HotpotQA are relatively less challenging for modern LLM agents, typically involving only a few hops where context management is not the main bottleneck. In contrast, BrowseComp is increasingly used in long-context agent research and places stronger demands on long-horizon reasoning and context management. That said, we agree that including multi-hop QA benchmarks would improve comparability. Due to the limited rebuttal time required for additional training and evaluation, we will run experiments on these QA benchmarks and include the results in the final version.
>
> ---
>
> > **The training of the proposed method might generalize to Qwen/LLaMA series as the Seed-OSS is not popular in research literature and remains unknown for comparison fairness.**
>
> Thanks for your suggestion! We chose Seed-OSS-36B as the base model because, at the time of our experiments, other available alternatives such as the Qwen3 series achieved relatively low scores on SWE-Bench tasks. With more strong base models now available, we plan to include results with additional models in the final version.
>
> Regarding comparison fairness, our current comparisons are conducted using the same base model and training data across methods to ensure a fair comparison. In addition, the Seed-OSS-36B architecture is same as Qwen2.5, which could make future comparisons with other models straightforward.
>
> ---
>
> > **The implementation of the GRPO RL process with changing trajectories (e.g., varied prefix contex) should be fully described in details, since this would affect the optimization of GRPO.**
>
> Thanks for the suggestion. As discussed in Appendix A.1, for multi-branch agent rollouts, instead of concatenating all sub-trajectories into a single sequence, we keep them as separate causally conditioned sequences and optimize them using a multi-sequence GRPO variant, with advantages calculated as in Eq. (6). We will include more implementation details of the RL process in the final version and will open-source our code for reproducibility.
>
> ---
>
> > **More training and validation dynamics such as the number of folding actions, the context window size, and the number of samples (not the trajectory-level but the actual samples involved in policy gradient update) are highly encouraged to provide with in-depth discussion.**
>
> The number of folding actions is the same as the number of branches created, as shown in Figure 4. The context size is fixed at 32K for both the main thread and branches during training. The number of samples in a policy update step corresponds to all sequences generated from a batch, which is approximately the average number of branches created multiplied by the number of prompts. We add padding sequences to ensure the total number of sequences is divisible by the batch size, but since we enable padding removal, these do not contribute to training cost. We will add more clarification and discussion in the paper.
>
> ---
>
> > **The following work on RLVR for agentic scenarios can be considered to add into the related work**
>
> Thanks for the pointer! We will add discussion to these work in our related work section.
>
> ---
>
> **Thank you again for your thoughtful feedback! We hope these clarifications will resolve your concerns.**

---

> > ### Author Rebuttal · Reviewer_dD92 · 2026-04-03
> >
> > The authors promise that codes will be released for details and new further experiments would be added into the final version to improve the manuscript. The reviewer hopes the community can benefit from such open-sourced projects.

---

### Official Review · Reviewer_Fb92 · 2026-03-07

**Soundness:** 3
**Presentation:** 2
**Significance:** 3
**Originality:** 3
**Overall Recommendation:** 5
**Confidence:** 4

**Summary:**

This paper proposes Context Folding, a mechanism that enables LLM agents to actively manage their working context on long-horizon tasks. The agent is equipped with two tools: branch(description, prompt) to spawn a sub-trajectory for a subtask in a separate context, and return(message) to fold that sub-trajectory back by keeping only a concise summary. To train this behavior end-to-end, the authors propose FoldGRPO, which augments standard GRPO with (i) dynamic context folding during rollouts and (ii) three token-level process rewards — an Unfolded Token Penalty, an Out-of-Scope Penalty (judged by GPT-5-nano), and a Failure Penalty. Evaluated on BrowseComp-Plus (deep research) and SWE-Bench Verified (agentic coding) with a Seed-OSS-36B model, the Folding Agent achieves 62.0% and 58.0% pass@1 respectively, matching or outperforming a 327K-context ReAct baseline while using only 32K active context (up to 10 branches). The RL training provides +20% and +8.8% absolute improvements.

**Compliance With Llm Reviewing Policy:**

Affirmed.

**Final Justification:**

The rebuttal addresses my main concerns regarding novelty, the nested branching limitation, and the modest SWE-Bench gains, with well-reasoned justifications. I raise my score to 5, as the core framework is sound and practically impactful, and the authors demonstrate a clear understanding of the method's scope and limitations.

**Key Questions For Authors:**

- Q1: Have you experimented with different thresholds for the Unfolded Token Penalty (e.g., 30%, 70%)? How sensitive is the method to this choice?

- Q2: What is the accuracy of GPT-5-nano as the out-of-scope judge? Have you measured its precision/recall? What happens if you replace it with a smaller or open-source model?

- Q3: The SWE-Bench gains are modest (+2.8%). Do you have a hypothesis for why context folding helps less on coding tasks than on deep research? Is it because SWE tasks are less context-bound, or because the folding summaries lose critical code details?


- Q4: Why was nested branching disabled? Was it tried and found to hurt performance, or was it excluded a priori? Can you discuss the challenges of multi-level folding?

- Q5: How does the method perform with other base models (e.g., Qwen3, LLaMA)? Is the benefit of context folding model-specific?

- Q6: Table 1 shows the Folding Agent uses 19.2 tool calls on BrowseComp (vs. 10.2 for 327K ReAct+RL) and 96.5 on SWE (vs. 55.4). Does this higher tool-call count translate to higher inference cost despite the smaller context?

**Limitations:**

yes

**Strengths And Weaknesses:**

## Strengths

- **S1**: The paper precisely identifies the core bottleneck of long-horizon agents — linearly growing context — and proposes an elegant abstraction: branch/return as learnable tools. The formulation (Eq. 3-4, Section 3) is concise and the context manager F(·) is well-defined. Figure 1 immediately conveys the intuition.

- **S2**: The method achieves remarkable context efficiency — 90% context compression (main thread ~8K out of 100K+ total tokens, Table 2) while matching or exceeding a 327K ReAct baseline. The +20% improvement on BrowseComp-Plus via RL (Table 1) is substantial. The training efficiency gain (1.5x over 327K ReAct, Figure 6) is also practically meaningful.

- **S3**: The paper provides extensive analysis: (i) ablation of FoldGRPO vs vanilla GRPO (Table 2) clearly showing the process rewards are critical; (ii) per-difficulty breakdown (Figure 3) showing gains concentrate on medium/hard instances; (iii) test-time scaling to 50 combined questions (Figure 5 right) demonstrating length generalization well beyond training distribution (10 branches → 32.6 branches); (iv) training dynamics (Figure 4) showing the agent learns to use more tools and branches over time.

- **S4**: The KV-cache rollback for inference efficiency (Section 3), shared context prefix between main and sub-agents (Section 4.3), and asynchronous rollout to mitigate training bubbles (Appendix A.2) are well-thought-out engineering contributions.


## Weaknesses

- **W1**: Context Folding is essentially a learned, single-model version of multi-agent sub-task delegation. The paper acknowledges this in Section 4.3 but understates the similarity. The `branch` action creates a sub-agent with a separate context; `return` collects the result — this is functionally equivalent to how systems like Claude Code's sub-agents or Anthropic's multi-agent research system work. The genuine novelty lies in (i) making this end-to-end trainable and (ii) the process reward design, but the mechanism itself is not fundamentally new. The paper would benefit from a more honest discussion of what is truly novel vs. what is a formalization of existing practice.


- **W2**: The Out-of-Scope Penalty uses GPT-5-nano to judge whether a branch stayed on-topic (Section 4.2). This introduces (i) a dependency on a closed-source model during training, making reproducibility difficult; (ii) potential noise in the reward signal — the paper acknowledges "future work may design more robust judgers" (Appendix E); and (iii) cost and latency overhead not quantified. How sensitive is the method to the quality of this judge? The ablation (Figure 12) shows removing the scope penalty degrades scope accuracy, but no analysis explores what happens when the judge is noisy or biased.


- **W3**: The paper explicitly disables nested branching (Section 3, "creating new branches is disabled while in execution state"). This is a significant limitation: many real-world tasks have naturally hierarchical structure (e.g., debugging requires exploring a file, which requires understanding a function, which requires reading related code). The paper mentions "multi-layer context folding" as future work (Section 8) but doesn't discuss the design challenges or why this was excluded beyond preventing "nested complexity."

- **W4**: On SWE-Bench Verified, the Folding Agent (58.0%) only marginally outperforms the 327K ReAct+RL baseline (57.4%, +2.8pp after RL). This raises the question of whether context folding provides significant benefit when the base model is already strong at coding tasks, or whether the gain is primarily from BrowseComp-style information retrieval tasks.

- **W5**: "Appendix ??" appears in Section 5.1, indicating an unresolved cross-reference. Table 2 caption uses "FoldPO" instead of "FoldGRPO" — inconsistent naming.

- **W6**: The Unfolded Token Penalty activates when main thread context exceeds 50% of the limit (Section 4.2). Why 50%? No ablation or analysis is provided for this threshold. It seems like an important design choice that could significantly affect behavior.

---

> ### Author Rebuttal · Authors · 2026-03-31
>
> Thank you for the detailed summary and for recognizing the novelty, context efficiency, and extensive analysis of our work! We would like to address your concerns as follows:
>
> ---
>
> > **W1**
>
> Thank you for this observation. We agree that context folding can be viewed as a specific instance of the broader multi-agent or subtask delegation paradigm, and we discuss this connection in Section 4.3. Our contribution is not delegation itself, but integrating branching, context compression, and credit assignment into a single learned framework for long-horizon tasks.
>
> Our approach differs from typical multi-agent systems in that roles are not fixed as separate models but dynamically created through branching within a single model, context is not passed verbatim but selectively folded via learned summarization, and the entire process is trained end-to-end so the model learns how to allocate context and compress information to improve long-horizon performance. **We will clarify this relationship to prior multi-agent work more clearly in the introduction.**
>
> ---
>
> > **W2**
>
> Thank you for this helpful suggestion. We use GPT-5-nano as the scope judge because it is inexpensive, fast, and stable compared to self-hosted models; the total cost is typically under $10 per training job, so the overhead in cost and latency is small relative to overall training. Regarding judge quality and noise, as discussed in Appendix E, we improved reliability through manual inspection and iterative prompt refinement until no major systematic errors were observed.
>
> ---
>
> > **W3 & Q4**
>
> Thank you for raising this important point. We agree that nested branching and hierarchical folding are promising directions, but we intentionally focus on single-layer folding for two reasons. First, in the agentic tasks we study, **most tokens are spent on tool-based execution rather than high-level planning**, so single-layer folding already captures most of the context-management benefit; planning branches typically use far fewer tokens, and if a subtask exceeds the context limit, the agent can summarize progress and spawn a new branch to continue. Second, **the base models still struggle with nested control flow,** often forgetting execution state, failing to return from sub-branches, or losing track of branch context; although we partially mitigate this with an unfolded token penalty, such errors compound in multi-layer settings and hurt performance, likely because LLMs are primarily pretrained on sequential text rather than hierarchical execution traces. We agree hierarchical folding is an important future direction and will discuss the design challenges and trade-offs in the final version.
>
> ---
>
> > **W4**
>
> Thank you for raising this point. We believe the key factor is effective context length rather than task domain. Many SWE-Bench tasks fit within a ReAct-style agent’s context window, so context management provides only limited benefit. In contrast, BrowseComp requires much longer context, where context folding becomes more important and leads to larger gains. In longer-horizon coding scenarios, such as repository-wide refactoring, context management remains important.
>
> ---
>
> > **W5**
>
> Thank you for catching this. We will fix the appendix reference in Section 5.1 and correct the naming inconsistency in Table 2.
>
> ---
>
> > **W6 & Q1**
>
> Thank you for this question. We use a 50% threshold because the model has a 32K context window and branches share the same budget, effectively reserving ~16K tokens for branch execution. A smaller buffer may cause premature truncation, while ~16K provided a reasonable balance between preserving main-thread context and allowing sufficient space for branch computation.
>
> ---
>
> > **Q2**
>
> We do not build a specific test set for this judge. As described in Appendix E, we improve the judge through manual inspection on sampled data.
>
> ---
>
> > **Q3**
>
> Thank you for this question. Compared to BrowseComp, SWE-Bench is generally less context-bound and many tasks can be solved within the context window of the ReAct agent, so context folding provides smaller gains. We therefore believe the difference comes from the characteristics of the benchmarks rather than an inherent difference between coding tasks and deep research tasks.
>
> ---
>
> > **Q5**
>
> Qwen3 and LLaMA show low base performance on SWE-Bench, where results are primarily limited by coding ability rather than context length. Among evaluated models, Seed-OSS performed best for ReAct agents, so we focused subsequent experiments on it. We agree that evaluating additional base models would be valuable, and we plan to include newer models such as Qwen3.5 in the final version.
>
> ---
>
> > **Q6**
>
> We discuss rollout and training latency in Figure 6 and Section 6. Overall, our method has lower average rollout/training latency than ReAct.
>
> ---
>
> **Thank you again for your thoughtful feedback! We hope these clarifications will resolve your concerns.**

---

> > ### Author Rebuttal · Reviewer_Fb92 · 2026-04-03
> >
> > The authors address the weaknesses and the questions. And I raise the overall score 4->5.

---

### Official Review · Reviewer_uj6s · 2026-03-10

**Soundness:** 3
**Presentation:** 3
**Significance:** 3
**Originality:** 3
**Overall Recommendation:** 5
**Confidence:** 3

**Summary:**

The paper proposes context-folding, a framework to enable LLMs to branch and assign subtask efficiently to other instances of themselves, reducing context length and improving performance. They also propose FoldGRPO, a variant of GRPO to make this behaviour learnable.

**Compliance With Llm Reviewing Policy:**

Affirmed.

**Final Justification:**

The paper has strong contributions and has good potential impact. I have increased my score from 4 to 5.

**Key Questions For Authors:**

How does it perform on some actual long-horizon interactive tasks, with hundreds of timesteps of interaction with an evolving environments, where planning and more careful memory management are required? I think the paper is accept-worthy as is, but further experiments could take it to a stronger score.

**Limitations:**

yes

**Strengths And Weaknesses:**

**Strengths**:

The paper tackles a very important challenge, and is quite timely. Results seem strong across the tested tasks, and work has the potential to become highly impactful.

The paper is well written, clear and easy to follow.

**Weaknesses**:

- These are not really what I would consider "long-horizon" tasks. They are "atomic-tasks" that agents can better solve through tool use and by taking an arbitrary number of reasoning and tool-usage steps to solve. How does context-folding and FoldGRPO behave in truly long-horizon tasks where an agent might go through hundreds of repeated actions-observation interactions with a persistent environments that evolves as the agent acts? Would important nuances be lost in some of the folds? In those cases, perhaps some summarization techniques would likely be more helpful, or do the returns automatically learn how to do that? I'm thinking of long-horizon environments like OSWorld [1], ALFWorld [2] or BALROG [3]. Adding more environments would make the results more general.
- Furthermore, both SWEbench and DeepResearch are somewhat similar static tasks, and reward sequential subtask executions. Nothing really wrong can happen within these tasks these environments, so there doesn't seem much of a need to adapt to failure cases. How would context folding work in tasks that don't reward this structure as much? Tasks where this hierarchical decomposition is not necessarily straightforward? Some of the tasks proposed in the previous bullet point could be good for this.

Editorial nit: missing appendix reference line 223


[1] Xie, Tianbao, et al. "Osworld: Benchmarking multimodal agents for open-ended tasks in real computer environments." Advances in Neural Information Processing Systems 37 (2024): 52040-52094.

[2] Shridhar, Mohit, et al. "Alfworld: Aligning text and embodied environments for interactive learning." arXiv preprint arXiv:2010.03768 (2020).

[3] Paglieri, Davide, et al. "Balrog: Benchmarking agentic llm and vlm reasoning on games." arXiv preprint arXiv:2411.13543 (2024).

---

> ### Author Rebuttal · Authors · 2026-03-31
>
> Thanks for the positive feedback and thoughtful questions! We are glad that you find the problem important, the results strong, and the paper clear. We would like to address your concerns as follows:
>
> ---
>
> > **Definition of long-horizon tasks**
>
> We would like to first clarify our use of the term "long-horizon." In this work, we use it to refer to tasks that require an effective context size far exceeding a model’s native capacity, or extended multi-step interactions that would take a human expert substantial time to complete.  Under this definition, SWE-bench and BrowseComp are relevant examples: solving them often requires processing 100K+ tokens, and agents may perform hundreds to thousands of reasoning and tool-use steps during execution. They are also widely used benchmarks for real-world agent applications such as coding and research workflows.
>
> ---
>
> > **Capturing nuances and mitigating information loss**
>
> We agree that an important question is whether context folding might discard subtle but critical information. Addressing this challenge is the key motivations behind our method. Rather than relying on a fixed heuristic, FoldGRPO is trained end-to-end so that the model learns how to compress branch information while preserving what is most useful for downstream decision-making. In other words, through return optimization, the method automatically learns effective branch summarization—retaining the nuances that matter for future reasoning.
>
> ---
>
> > **Adapting to failure cases and less structured tasks**
>
> Regarding the need to adapt to failures: while SWE-bench and BrowseComp have clear end goals, the trajectory to reach them is rarely straightforward. Agents frequently encounter failure cases such as syntax errors, failed unit tests, or dead-end web navigations. In these cases, context folding can help by compressing failed attempts into short summaries rather than keeping long error logs in the context. This allows the agent to efficiently adapt, backtrack, and explore alternative strategies without losing sight of the broader objective—a mechanism we expect to translate well to environments where hierarchical decomposition is not natively rewarded.
>
> ---
>
> > **Evaluation on additional environments**
>
> We appreciate the suggestion to evaluate on additional interactive environments such as OSWorld, ALFWorld, and BALROG, and agree these are valuable directions. We note that ALFWorld is typically solvable within a much smaller context window (often around 4K tokens) and therefore places less emphasis on context management. OSWorld and BALROG are highly compelling, but since they are multimodal and require non-trivial additional infrastructure, this is beyond the scope of the rebuttal period. We agree they would be valuable for future evaluation.
>
> ---
>
> **Thank you again for your thoughtful feedback! We hope these clarifications will resolve your concerns.**

---

> > ### Author Rebuttal · Reviewer_uj6s · 2026-04-01
> >
> > My concerns are mostly resolved, and I'm increasing the score from 4 to 5.

---

### Official Review · Reviewer_FuJC · 2026-03-12

**Soundness:** 3
**Presentation:** 4
**Significance:** 3
**Originality:** 3
**Overall Recommendation:** 5
**Confidence:** 4

**Summary:**

In this paper, the authors present a novel approach for managing long contexts in agentic setups. The authors propose a method called "Context Folding" which relies on branching of the context with subsequent folding and summarization. To foster the proposed scheme with branching, authors introduce FoldGRPO which mainly rely on the careful design of the reward function. The authors compare the proposed method with summarization and long context baselines on BrowseComp-Plus and SWE-Bench verified datasets. The results show significant improvements in metrics compared to the baselines, demonstrating the effectiveness of the proposed approach.

**Compliance With Llm Reviewing Policy:**

Affirmed.

**Final Justification:**

A solid paper that proposes a novel method for managing long contexts. The rebuttal addressed all main concerns and reinforced the decision of scoring the work with 5 (Accept)

**Key Questions For Authors:**

Questions:
1. For table 2, Main Len is above 32k in contrast to Table 1, which constrains the main length to 32k, how is it possible?
2. How the wall-clock time of FoldGRPO compares to the baselines? Is it more computationally expensive compared to baselines?
3. How to control the number of branches in the proposed method?

**Limitations:**

yes

**Strengths And Weaknesses:**

Strengths:
- strong baselines considered for comparison
- significant reduction of the context length, which may be beneficial both from the model hard limits of max context length and memory consuption perspectives
- siginificant improvements of metrics in compared to baselines
- generalization to unseen longer number of branches

Weaknesses:
- the reward function is highly engineered and possibly ad-hoc, which may limit the generalizability of the approach
- single model is used for training and evaluation of FoldGRPO
- from the number of tool-calls in the Table 1, it might be possible that the model trained with FoldGRPO more computationally expensive compared to baselines

---

> ### Author Rebuttal · Authors · 2026-03-31
>
> Thanks for your thoughtful and constructive feedback! We are glad that you recognize the strong baselines, context reduction benefits, and generalization ability of our approach. We appreciate your concerns regarding the engineered reward design, single-model evaluation, and computational cost, and we address these points below:
>
> ---
>
> > **the reward function is highly engineered and possibly ad-hoc, which may limit the generalizability of the approach**
>
>
> Thank you for raising this point. We agree that our current reward function involves specific engineering choices, which were necessary to stabilize training and prevent reward hacking. However, we believe the underlying mechanisms—specifically, incorporating process rewards into GRPO and penalizing out-of-scope or premature unfold behaviors—are broadly applicable. These components are designed to address fundamental LLM limitations in context management across different tasks, rather than being tied to a specific domain. That said, we completely agree with your assessment that exploring simpler, more elegant reward designs is a valuable direction for future work.
>
> ---
>
> > **single model is used for training and evaluation of FoldGRPO**
>
> Thanks for pointing this out. We chose Seed-OSS-36B as the base model because, at the time of our experiments, other available alternatives such as the Qwen3 series achieved relatively low scores on SWE-Bench tasks. With more strong base models now available, we plan to include results with additional models in the final version.
>
> ---
>
> > **from the number of tool-calls in the Table 1, it might be possible that the model trained with FoldGRPO more computationally expensive compared to baselines**
>
> We appreciate the reviewer raising this point. While our model does generate more output tokens and tool calls during RL training, this increase is primarily concentrated within the hard subset of tasks. For easier instances, the computational overhead increases only marginally, as demonstrated in Figure 4. We argue that successfully solving challenging agent tasks inherently requires additional computation—a fundamental limitation of standard ReAct models, which struggle to effectively leverage longer horizons. Furthermore, despite the increased tool usage, Figure 6 shows that our approach actually achieves lower overall latency because it actively maintains a concise context. We agree that balancing effectiveness with efficiency is crucial, and we view the development of more efficient long-horizon scaling methods as a promising direction for future work.
>
> ---
>
> > **For table 2, Main Len is above 32k in contrast to Table 1, which constrains the main length to 32k, how is it possible?**
>
> Thank you for catching this discrepancy. This anomaly was due to a logging artifact in our length calculation. When the agent receives an excessively long tool output (such as a massive code file or execution log), the raw output was temporarily appended to the chat history and counted toward the "Main Len" statistic before our system truncated it to the 32k maximum limit. The model itself never actually processed a context larger than 32k. We appreciate you pointing this out; we will clarify this detail and update the results with the correct post-truncation lengths in the final version.
>
> ---
>
> > **How the wall-clock time of FoldGRPO compares to the baselines? Is it more computationally expensive compared to baselines?**
>
> We discuss training and rollout latency in Figure 6 and the Model Efficiency section in Section 6. Overall, our approach has lower average rollout / training latency than the ReAct agent.
>
> ---
>
> > **How to control the number of branches in the proposed method?**
>
> When the model reaches the branch number limit, the create-branch tool returns the message: “You’ve already reached the limit of {len(branches)} branch calls. Continue working independently.”
>
> ---
>
> **Thank you again for your thoughtful feedback! We hope these clarifications will resolve your concerns.**

---

> > ### Author Rebuttal · Reviewer_FuJC · 2026-04-04
> >
> > The concerns raised in the review are fully resolved, therefore, I keep the score positive.

---

### Decision · Program_Chairs · 2026-04-30

**Decision:**

Accept (regular)

**Comment:**

This paper proposes Context Folding, a framework that lets an agent actively manage its working context by branching into sub-trajectories and folding them back into concise summaries, together with an RL training method that makes this behavior learnable. I recommend acceptance. The main strengths are clear. The paper addresses an important bottleneck for long-horizon agents, the method is well motivated and practically meaningful, and the empirical results are strong. In particular, the paper shows that the agent can maintain a much smaller active context while remaining competitive with or better than strong long-context baselines. The reviewers also found the analysis section thorough and the presentation generally strong.

The rebuttal addressed the main concerns. In particular, it clarified the logging artifact around context length, explained the efficiency tradeoff, clarified the relation to multi-agent delegation, and discussed why nested branching was not included in the current version. I agree with the reviewers that the core contribution is not delegation itself, but turning context branching and folding into a learned, end-to-end trainable mechanism for context management.

There are still limitations. The reward design is somewhat engineered, the evaluation is concentrated on two task families, and broader validation on more interactive long-horizon environments would strengthen the paper further. But these are limitations of scope rather than reasons to reject.